# Accuracy Assessment of Joint Angles Estimated from 2D and 3D Camera Measurements

**DOI:** 10.3390/s22051729

**Published:** 2022-02-23

**Authors:** Izaak Van Crombrugge, Seppe Sels, Bart Ribbens, Gunther Steenackers, Rudi Penne, Steve Vanlanduit

**Affiliations:** Faculty of Applied Engineering Department Electromechanics, Universiteit Antwerpen, Groenenborgerlaan 171, 2020 Antwerpen, Belgium; izaak.vancrombrugge@uantwerpen.be (I.V.C.); seppe.sels@uantwerpen.be (S.S.); bart.ribbens@uantwerpen.be (B.R.); gunther.steenackers@uantwerpen.be (G.S.); rudi.penne@uantwerpen.be (R.P.)

**Keywords:** skeletonization, triangulation, range cameras, multi-camera, ergonomic evaluation, REBA

## Abstract

To automatically evaluate the ergonomics of workers, 3D skeletons are needed. Most ergonomic assessment methods, like REBA, are based on the different 3D joint angles. Thanks to the huge amount of training data, 2D skeleton detectors have become very accurate. In this work, we test three methods to calculate 3D skeletons from 2D detections: using the depth from a single RealSense range camera, triangulating the joints using multiple cameras, and combining the triangulation of multiple camera pairs. We tested the methods using recordings of a person doing different assembly tasks. We compared the resulting joint angles to the ground truth of a VICON marker-based tracking system. The resulting RMS angle error for the triangulation methods is between 12° and 16°, showing that they are accurate enough to calculate a useful ergonomic score from.

## 1. Introduction

The importance of good ergonomics for workers is hard to overstate. Bad body postures can lead to musculoskeletal disorders (MSDs) which have a large impact on both the workers lives and the economy. To prevent MSDs, the ergonomics must be monitored so that problems are detected early. Most ergonomic assessment methods are done manually. But they could be also performed automatically, based on the joint angles of the 3D skeletons.

Three of the most used [1] ergonomic assessment methods—RULA [2], REBA [3] and OWAS [4]—can be calculated largely based on the joint angles. We will use REBA (Rapid Entire Body Assessment) as an example use case because it demonstrates the use of many different joint angles. To correctly calculate the joint angles, they must be calculated in 3D space because they are distorted when they are projected onto a 2D image. Therefore, 3D skeletons are needed.

In this work, we calculate 3D skeletons based on camera measurements. In our approach, we first determine the 2D skeletons in the camera images using a convolutional neural network, more specifically Detectron2 [5]. The 2D skeletons can then be converted into 3D skeletons in different ways:Using the depth image from a single RealSense range camera.Triangulating the joints using multiple cameras.Combining the triangulation of multiple camera pairs.

The first method has the advantage that there would be no need for camera overlap. However, the second and third method could be used with standard 2D cameras (no depth perception required). We will test each method and evaluate their accuracy.

2D skeleton detectors—like the one used in the first step—have an abundance of training data as there are many large and varied annotated single-view datasets available (e.g., COCO [6] and MPII [7]).As a result they can achieve very high accuracy and robustness. Multi-view skeleton detectors are trained on multi-view datasets, which are generally smaller in size and recorded in controlled environments where both the background and the subjects have limited visual diversity [8].

We describe and test 3D skeletonization techniques based on the depth data of a single RealSense or on the triangulation between two or three cameras. We evaluate the resulting joint angles by comparing them to an accurate VICON marker-based tracking system. Those joint angles are elemental to the calculation of ergonomic scores, e.g., with the REBA [3] method. To our knowledge, this is the first time these methods are systematically compared to a marker-based method.

The joint angle errors are smallest when combining triangulations and largest when using the RealSense depth image. A notable result is that the three *key angles*—angles that have the most effect on the ergonomic score: trunk, shoulder and knee flexion—are the most accurately reconstructed angles.

## 2. Related Work

Two decades ago, the research field of computer vision-based human motion capture was still in an early stage of development [9]. While high accuracy pose tracking techniques existed, the hardware was expensive and the recording procedures were elaborate. As a result they were in practice only used by well-equipped labs and Hollywood studios. The markerless methods depended on many assumptions about the environment and subject. The main technique to achieve depth images was stereo vision, which suffered from low frame rates due to the high computing cost.

The introduction of depth sensors, specifically the low-cost Microsoft Kinect V1 in 2010 and Kinect V2 in 2013 boosted the research in pose tracking. (The correct names of the first and second generation Microsoft Kinect sensors are *Kinect for Xbox 360* and *Kinect for Xbox One*. For clarity and conciseness, they are commonly called Kinect V1 and Kinect V2 respectively). More datasets with depth and annotated keypoints became available [10], facilitating further research in the field. The Kinect V1 algorithm uses trained classifiers to segment different body parts in the depth image and then derive the keypoints from them [11].

Detection and segmentation improved quickly over the years with the improvement of convolutional neural networks. Cao et al. made a big step forward in speed and accuracy of keypoint detection with their OpenPose detector [12]. A similar keypoint detector is implemented in the open source neural network platform Detectron2 [5]. This is the detector we use to detect 2D keypoints.

In the field of 3D human pose estimation there is much research on single view 3D estimation [13,14,15]. They have the advantage that they can be used in many applications, as only a simple camera is needed. This flexibility, however, comes at the cost of significantly lower accuracy compared to multi-camera techniques. Because 2D images have too much ambiguity to robustly estimate a person’s 3D pose, we will use multiple views in our application.

We are not the first to triangulate 2D detections into 3D keypoints. Belagiannis et al. [16] and Amin et al. [17] first detect body parts followed by algebraic triangulation with calibrated cameras. They use the N-view triangulation method of Hartley and Zisserman [18]. We used a different method [19], it offers the geometric triangulation error δ directly. We use δ when combining triangulations, it is a more appropriate error metric than the reprojection error. Dong et al. [20] first detect 2D poses using a cascaded pyramid network [21] and convert them to 3D using triangulation and also a pictorial structure model.

Our work differs from the existing literature that uses triangulation in several ways:We use simple algebraic triangulation, where many methods use neural networks for triangulation [22] or require training of weights [23].Our method can be seen as a simplified version of [24]. But in our algorithm the two parts—detection and triangulation—are completely independent, so the detector can be upgraded to the best state of the art.Most methods assume no extrinsic calibration to allow for moving cameras [25,26]. We have fewer uncertainties by using calibrated static cameras.

Not many state-of-the-art papers analyze the joint angle errors. The cited manuscripts use one of following two error metrics:PCP (percentage of correct parts) [27] or the closely related PCK (percentage of correct keypoints) [28] and PCKh [7]. It expresses how many of the keypoints were correctly detected.MPJPE (mean per joint position error) is the mean Euclidean distance between all keypoints and their respective ground truth locations.

Similar to our work, Choppin et al. [29] determined the joint angle errors of two 3D skeletonization methods, compared to a marker-based tracking system. But they only evaluated the performance of a Kinect V1 and did not use multiple cameras.

## 3. Algorithms for 3D Skeleton Reconstruction

Our three techniques for 3D keypoint calculation are laid out in Figure 1.

### 3.1. 2D Skeletonization

Detectron2 [5] is used in post-processing to determine the 2D skeletons based on the visual images. It detects the position of 18 keypoints as defined in the COCO standard [30] along with their confidence value. This detection is done on each frame for all three cameras. The used skeleton model is shown in Figure 2.

### 3.2. 3D Coordinates from RealSense Depth

The first method to convert the 2D keypoints into 3D, is the naive approach. We reconstruct the 3D skeleton based on the data of a single RealSense sensor. Using a single sensor has the advantage that the fields of view do not need to overlap.

For each 2D keypoint, the corresponding depth is taken from the depth map that the RealSense camera generated. Because the keypoints are detected with sub-pixel resolution, they have non-integer coordinates. The depth is taken from the closest corresponding depth-pixel. By not using bilinear interpolation, we avoid introducing extra errors caused by pixels on edges or invalid pixels.

Because the keypoints have non-integer coordinates, the closest depth-pixel is chosen. From those depth values and the known intrinsic calibration of the RealSense we calculate the 3D coordinates of the keypoints.

### 3.3. 3D Coordinates from Triangulation

A second method to obtain 3D coordinates for the 2D keypoints is by using triangulation. This technique requires a full overlap of the cameras, but it does not need range cameras.

The poses of all cameras are known from the extrinsic calibration in Section 4.3. A 2D keypoint ($P_{1}$) is reprojected from its camera, according to the pinhole camera model as illustrated in Figure 3. This reprojection is a straight line extending from the camera. The corresponding 2D keypoint ($P_{2}$) is also reprojected from a second camera. Assuming perfect calibration and skeletonization, the two reprojection lines should intersect. But in reality they do not intersect. Instead, we find the 3D keypoint (*P*) where the distance ε between the two reprojection lines is smallest.

### 3.4. Combining Triangulations

Because three cameras are used, three different triangulations can be done: one for each pair of cameras. We can combine the three results into an superior result by a weighted average. For each point triangulated between camera *i* and camera *j*, the weight $W_{ij}$ is calculated as:(1)$$W_{ij}=\frac{c_{i}\cdot c_{j}}{{\left(\varepsilon_{ij}+\varepsilon_{m}\right)}^{2}}$$
with $\varepsilon_{ij}$ the reprojection error, the crossing distance between the reprojection lines as illustrated in Figure 3. And $c_{i}$ is the detection confidence for that keypoint in the Detectron2 result of camera *i*. To avoid extreme weights due to very small crossing distances, $\varepsilon_{m}$ is added. This is a *minimum crossing distance* with a chosen value of 1 mm.

We find the resulting point *P* by weighted average:(2)$$P=\frac{\sum W_{ij}\cdot P_{ij}}{\sum W_{ij}}$$

## 4. Experimental Setup

We will test the three proposed methods in a lab experiment to determine the accuracy of the calculated 3D skeletons. The measurement layout is described in Section 4.1 and the ground truth measurements in Section 4.2. We explain the used extrinsic calibration method in Section 4.3 and go over the recorded scenario in Section 4.4.

### 4.1. Used Hardware and Measurement Layout

The measurements are done using three RealSense range cameras, two D455’s and one D435. They perform on-chip stereo depth calculations, deliver aligned monochrome and depth images, and produce good results under various lighting conditions. Considering also the low price (less than €220 for a D455), they are a good candidate to be used in a real-world application.

Three RealSense cameras are placed pointing inward as shown in Figure 4. They are connected using a triggering cable, one camera is the master and two cameras are slave. The subject moves in the area between the cameras so that they are seen by all cameras.

The images of all left cameras are recorded, as well as the depth images, at 30 fps with a resolution of 1280 × 720. Depth is stored in 16-bit PNG files and the visual images are stored in 8-bit monochrome JPEGs.

### 4.2. VICON Ground Truth

The measurements are performed in the M²OCEAN lab of the University of Antwerp. It is equipped with 8 VICON MX T10 infrared cameras to track retroreflective markers. Markers are taped to a volunteering test subject precisely by a student physiotherapist. The marker trajectories are recorded at 100 fps. Using the VICON NEXUS software the trajectories are labeled manually and the skeleton model is inferred. Both the skeleton keypoints and the marker trajectories are then exported in a readable format.

To produce a useful ground truth, the data undergoes following steps:Downsample the data from 100 fps (VICON) to 30 fps (RealSense).Reorder the keypoints that the COCO and VICON skeleton model have in common.Reconstruct the remaining keypoints. For instance, the nose, eyes and ears of the COCO model are reconstructed from the four head markers.Synchronize the recording to the RealSense data by shifting it based on the cross-correlation of the *y*-coordinate.Find and apply a rigid transformation to match the world coordinate system.

### 4.3. Extrinsic Calibration

The method of extrinsic calibration can be chosen freely as long as an accurate method is used. We chose to use a variation of our calibration methods, but based on the fact that the three cameras see each other. It uses bundle adjustment with a the same cost function as in our earlier work [31,32]. We chose this method over our other methods because the fields of view have a large overlap and because it does not require a special procedure between recordings. So even if the cameras are moved between captures, the calibration can still be determined.

In post-processing, the camera centers are manually marked in a single image for each camera. To facilitate this, each camera has a recognizable white *monocle* mounted around the left camera, as shown in Figure 5. The relative pose of the cameras is found by an iterative optimization that minimizes the L2 reprojection error of the marked points.

The relative poses are known up to a scale factor. We calculate the scale as following average:(3)$$\mathrm{scale}=\langle \frac{d_{\mathrm{measured}}}{d_{\mathrm{calculated}}}\rangle$$
where $d_{\mathrm{measured}}$ is the distance from one camera center to another as it is measured by the RealSense camera and $d_{\mathrm{calculated}}$ is the distance between the calculated relative camera positions.

The accelerometer data of the D455 cameras is used to determine the world orientation with respect to the floor plane, as gravity causes a constant vertical acceleration. An estimated height of the first camera is provided manually.

### 4.4. Recorded Scenario

The goal of the experiment is to test the accuracy of the obtained joint angles. We chose to use a single human subject because the joint angles do not depend on the stature of the subject but rather on the adopted pose. Moreover, the used 2D skeletonization network has proven its performance on a varied population of people. The subject goes through a varied series of motions, resulting in challenging poses and a wide range of joint angles.

The tasks that the subject executed are chosen to produce poses that are relevant for workers in industry. The volunteer first picks up a nut and bolt and then fastens it to the structure at one of the four heights: 0.25 m, 1.15 m, 1.75 m, and 2.1 m. The test subject is free to perform the actions as they wish, so no specific pose is dictated. A number of resulting poses are shown in Figure 6a–g. A sequence of 3600 frames (120 s at 30 fps) is recorded and processed.

## 5. Experimental Results

Our application—calculating REBA scores—depends only on the angles between the skeleton joints. So accuracy of the joint angles is an important factor in our end result. Most joints have multiple degrees of freedom. From such joints, different joint angles are calculated (e.g., flexion, twist, bend), each with a single degree of freedom. From the skeleton, 11 different joint angles are calculated. The joint angles are listed on the horizontal axis in Figure 7 and their definition and calculation are detailed in Appendix A. Six of those angles are calculated for both the left and the right side. Because of symmetry, the left and right angles are combined in the results.

### 5.1. Angle Errors

There are different metrics to quantify how well the calculated angles match the true angles. A first one is to inspect the error, calculated as the difference between the results and the VICON ground truth. Figure 7 shows the box plots of those angle errors. Outliers—values more than 1.5 interquartile ranges above the upper quartile or below the lower quartile—are omitted from the chart for clarity. We report the RMS errors. This is a good metric because it gives more weight to large errors. The ergonomic score is based on the different joint angles falling within certain angle ranges. So different angles can result in the same score. As a result, small angle errors will have almost no impact on the resulting ergonomic score.

It is apparent that the more cameras that are used, the higher the resulting accuracy. The total RMS angle error based on the naive depth technique is quite high: 30°. The single and combined triangulation results are significantly better: 16° and 12° respectively.

### 5.2. Angle Correspondence

A second metric to evaluate the accuracy is to plot the results versus the ground truth. This way, the different kinds of errors are visualized. To quantify the goodness of fit in a single number, we report the *coefficient of determination* $R^{2}$ for each plot. It is calculated assuming the true relation between the compared values: y=x.
(4)$$R^{2}=1-\frac{\sum_{i}{\left(y_{i}-x_{i}\right)}^{2}}{\sum_{i}{\left(y_{i}-\bar{y}\right)}^{2}}$$

In Equation (Equation 4), $x_{i}$ and $y_{i}$ are the angles plotted on the horizontal and vertical axes respectively, $\bar{y}$ is the arithmetic mean. Note that by definition $R^{2}$ can also be negative.

In Figure 8 we compare the calculated angles to the *ground truth* of the VICON. We show a selection of 5 of the 11 angles, but the $R^{2}$ values for all angles are listed in Table 1. The angles calculated from the RealSense depth vary wildly from their true values. Only the *key angles*—flexions of the knee, shoulder and elbow—could be somewhat useful. For the other angles there is little correlation to the actual value.

The results improve when using the triangulation between two cameras. The accuracy of the key angles has increased considerably. Combining triangulations results in a further increase in those specific angle accuracies. The other joint angles do not seem to correspond well to the ground truth.

Table 1 confirms the trend that the accuracy increases from using only the depth to using combined triangulations. It also shows that none of the methods produces trunk twist, shoulder raise, and neck angles that correspond well to the VICON ground truth. This has two causes: on the one hand, our methods may be less accurate in the shoulder region. On the other hand, there seems to be a mismatch in the shoulder region between our skeleton model and the VICON one that could not be corrected in the skeleton conversion step. The VICON model has more flexible shoulders, while our skeleton detection produces more rigid shoulders.

## 6. Discussion

There are a number of known limitations when using our method to determine the joint angles for ergonomic evaluation like REBA. The skeleton detector we used produces skeletons in the COCO format, the hands and feet are not detected. So no ankle or wrist angle can be calculated. When needed, this can be solved by using a different skeleton detector that includes those joints. Our method only generates joint angles, so no other information is provided like e.g., force/load score, coupling score, and activity score that are used by REBA. When interpreting the automatically calculated scores, those factors must be taken into consideration.

One of the novelties of our work is that we evaluate the joint angles directly. This makes it hard to compare it directly to many other camera based methods, because the large majority uses position error metrics instead of angles. Even though joint angle errors are the most indicative metric for ergonomics, lost methods use MPJPE ([13,14,15,22,23,24,25]), PCP ([16,17,20,24,26,27,28]) or PCK ([13,15,22,28]).

As explained in Section 2, we evaluate the accuracy of the joint angles and not of the joint positions, as this is the most meaningful metric in the context of analyzing ergonomics. Choppin et al. [29] report a median RMS joint angle error of 12.6° for measurements with a Kinect V1. This seems comparable to our results, but was based on measurements where the subjects were facing the camera. As Plantard et al. [33] showed, the Kinect V1 is quite sensitive to the viewing angle: the accuracy decreases considerably with increasing viewing angles. Our results have higher RMS angle errors for the RealSense depth method (30°) and the single triangulation method (16°), but in our experiment we recorded the subject from all different angles.

Xu et al. [34] used a Kinect V2 to measure the shoulder joint angles of a person sitting in front of the camera and compared this to a motion tracking system. Their results had similar RMS errors—ranging between 8.5° and 38.5°—but with a static subject in a fixed position in front of the camera.

Abobakr et al. [35] estimated the 15 angles needed to calculate the RULA score directly in a deep convolutional network. On their real world measurements they had an RMS error over all joint angles of 4.85°. Their approach requires the training of a dedicated network, but offers a very good accuracy.

Li et al. [36] obtained a smaller RMS error of 4.77°. However, they used only images taken from an ideal viewpoint, so no random orientations were included. Their method require such viewing points—where the joint angles are parallel to the image plane—as they use 2D angles instead of 3D angles. The image should be taken by someone going around the workplace, taking pictures of different poses of interest, so this method is not intended to be used in a fully automatic system that observes workers over long periods of time.

## 7. Conclusions

In this paper, we constructed 3D skeletons from camera measurements and compared them to a VICON marker-based tracking system. The 2D skeleton keypoints are detected in the monochrome image. To convert those 2D skeletons to 3D skeletons, we have tested three methods: using the depth directly from the RealSense depth sensor, triangulating the points from two cameras, and combining the triangulations of all three cameras. We have evaluated the different method based on the accuracy of the joint angles, as those angles are essential for most applications like ergonomic evaluation.

Using the RealSense depth has the advantage that only a single sensor unit is needed. But it also has an important disadvantage: the accuracy of the 3D skeleton is limited. Compared to the VICON measurements, the RMS angle error is 30°. This will not be enough to calculate more than a rough estimate of the ergonomic score.

To better deal with occlusions, we combine two cameras to triangulate the corresponding keypoints. This requires an extra camera, but neither camera needs to be a depth sensor. The RMS angle error for the single triangulation is 16°. Those joint angles are accurate enough to obtain a good estimate of the ergonomic score.

Even with two cameras there can be still quite a number of occlusions. We therefore combine the three triangulations of the camera pairs. A weighted average is calculated based on the detection confidence of the 2D keypoint detector and on the triangulation quality. The combined triangulation method has an RMS angle error of only 12°. The results suggest that the accuracy could be improved further by adding even more cameras. However, the small gains in accuracy might not justify the additional hardware and increased demand for computing power for the skeletonization.

Not all joint angles have the same error. For all three methods the trunk, knee and shoulder flexion have a significantly higher accuracy than the other angles. Those *key angles* have a larger effect on the REBA score.

## Figures and Tables

**Figure 1 sensors-22-01729-f001:**
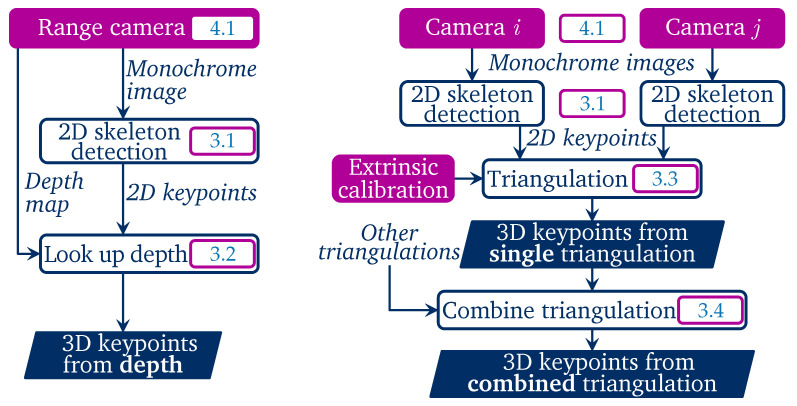
Block diagram of the 3D skeletonization algorithms with section numbers: based on range camera depth (**left**), based on single or combined triangulation (**right**).

**Figure 2 sensors-22-01729-f002:**
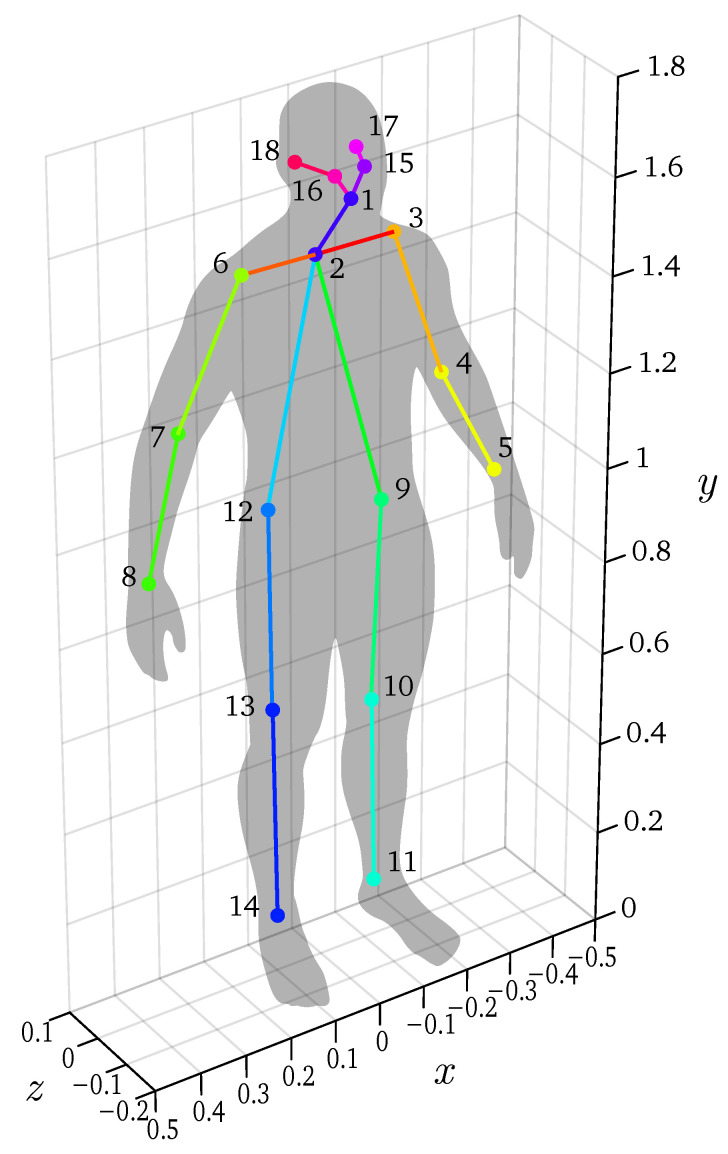
The used skeleton model.

**Figure 3 sensors-22-01729-f003:**
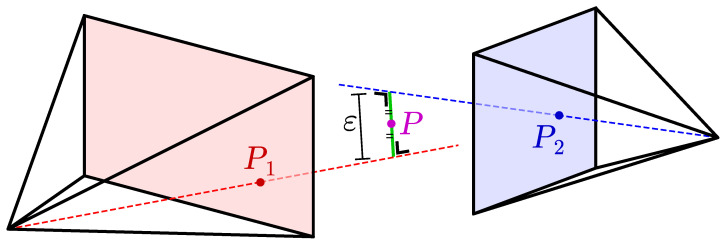
Illustration of the triangulation method for a single point.

**Figure 4 sensors-22-01729-f004:**
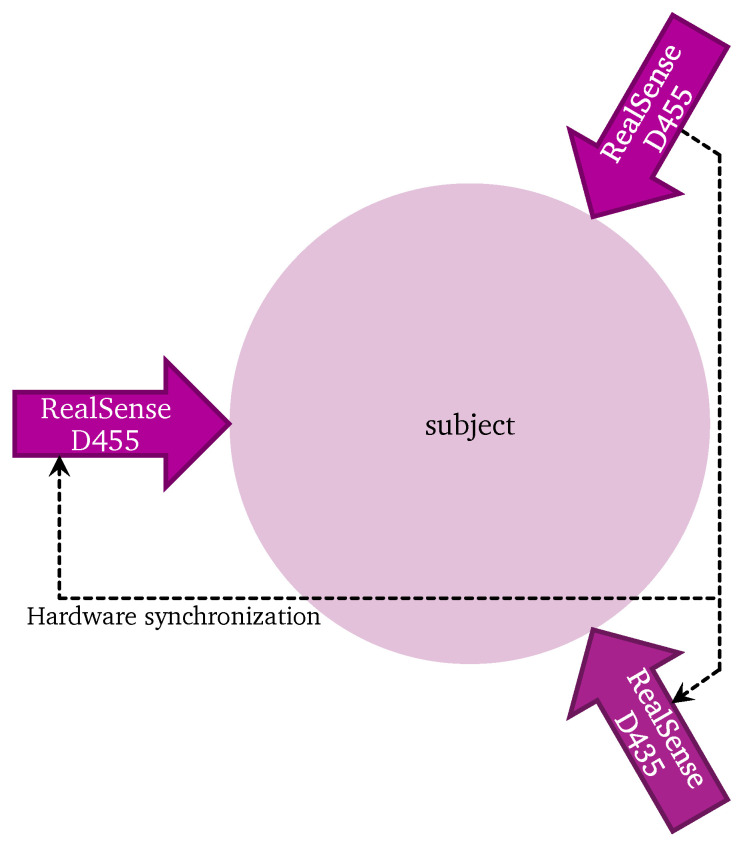
Physical setup of the experiment.

**Figure 5 sensors-22-01729-f005:**
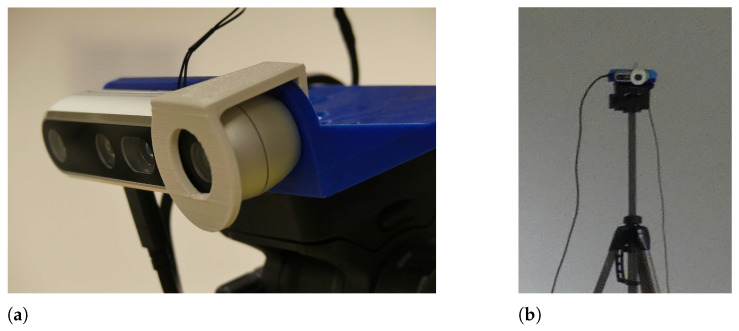
(**a**) The white monocle mounted on the camera. (**b**) The white ring is easily distinguishable, even in low resolution images.

**Figure 6 sensors-22-01729-f006:**
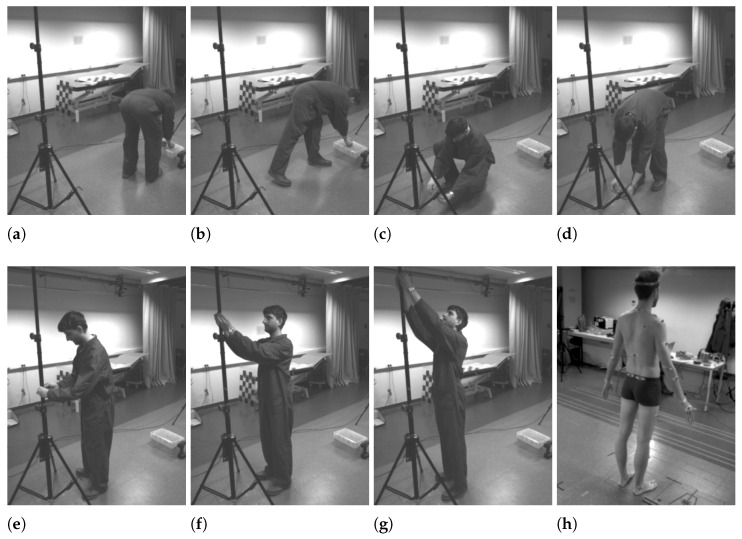
Different actions and poses of the subjects. Some actions can be performed in different poses. (**a**,**b**) Pick up a nut and bolt. (**c**,**d**) Fasten nut and bolt at 0.25 m, (**e**) at 1.15 m, (**f**) at 1.75 m, (**g**) at 2.1 m. (**h**) Subject with VICON markers.

**Figure 7 sensors-22-01729-f007:**
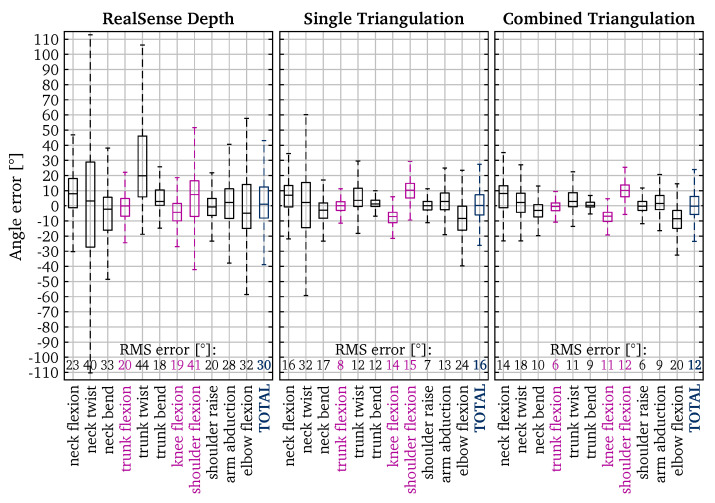
Error of calculated angles compared to the VICON ground truth. The key angles (joint angles with a high impact on the REBA score) are shown in fuchsia.

**Figure 8 sensors-22-01729-f008:**
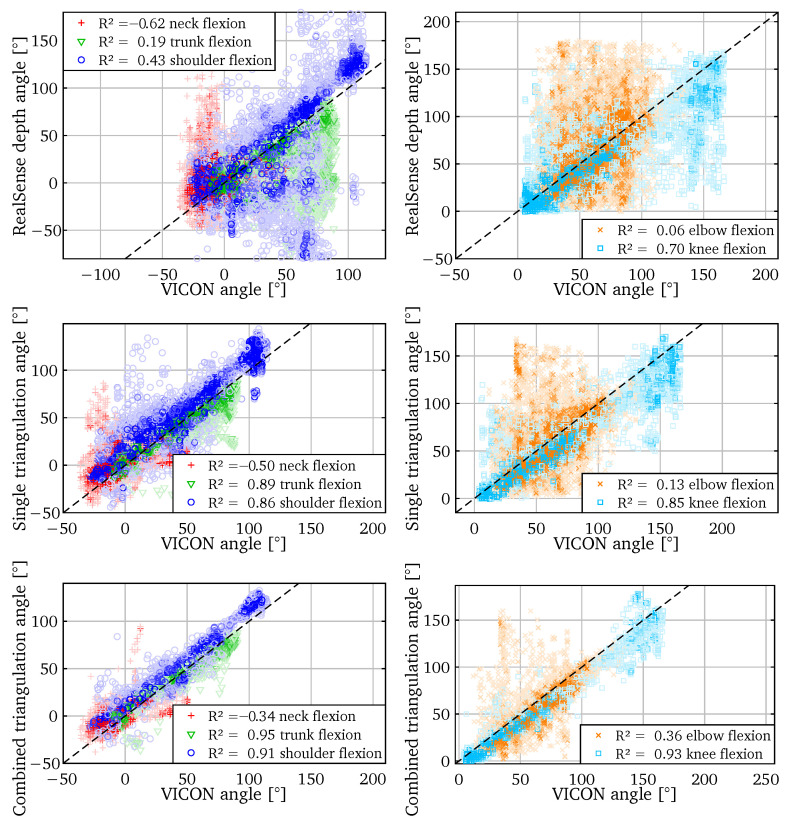
A selection of five joint angles compared to angles from the VICON data. From **top** to **bottom**: calculated from RealSense depth, single triangulation and combined triangulation.

**Table 1 sensors-22-01729-t001:** $R^{2}$ values for joint angles calculated using different methods. The three key angles and their values are shown in bold font.

	RealSense	Single	Combined
Joint Angle	Depth	Triangulation	Triangulation
neck flexion	–0.62	–0.50	–0.34
neck twist	–0.15	–0.26	–0.04
neck bend	–0.03	–0.03	–0.10
**trunk flexion**	** 0.19**	** 0.89**	** 0.95**
trunk twist	–0.93	–0.55	–0.35
trunk bend	0.06	0.31	0.45
**knee flexion**	** 0.70**	** 0.85**	** 0.93**
**shoulder flexion**	** 0.43**	** 0.86**	** 0.91**
shoulder raise	–0.02	–0.15	–0.11
arm abduction	0.00	0.04	0.28
elbow flexion	0.06	0.13	0.38

## Data Availability

The data presented in this study are available on request from the corresponding author. The data are not publicly available to protect the subjects privacy.

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
