# Peer review of "Accuracy Assessment of Joint Angles Estimated from 2D and 3D Camera Measurements"

_sensors, 2022, doi:10.3390/s22051729_

Round 1
Reviewer 1 Report
This is an interesting issue for the ergonomics practice. However, there are many important information not stated and led the difficulty to follow the main concept of the manuscript. After serious consideration, the quality of the manuscript with current form is not sufficient for academic report. For example, the reason for choose only one subject for the experiment and for the eight postures applied in the experiment are missing. The contribution and the importance of the study is still unclear. why need the 3D skeletons to replace 2D methods? The 2D methods (e.g. REBA with 2D angle) are reported as a good assessment with high accuracy and reliability. When using REBA for risk assessment, the wrist angle is one of the critical body parts for determination. However, the authors don't present the information about the wrist angle. The angle error (RMS) is about 30, 16 and 12 degree after comparison. It is quite large and need more clarification. Moreover, this paper lacks a discussion section to interpret and compare the experimental results with other studies. The title is not well represented the concept of the manuscript as well.
Reviewer 2 Report
The study discusses about 3D skeletons reconstructed based on 2D skeletonization. The manuscript is well written and interesting for biomechanics and ergonomics experts. Some additional information should improve the quality of the manuscript. My comments are as follows.
- What does REBA stand for? Please mention it at the beginning of the manuscript.
- RMS angle error is used for the evaluation. However, this parameter may indicate small value that make the method look so good. Could you add some information about the maximum errors? And, how does it compare with the maximum errors of VICON data?
- Could the method use more than 3 cameras? Do you think that use of more than 3 cameras will generate more accurate results? Discussion regarding limitations of the method will be welcomed.
- In related work, you mentioned about some existing methods proposed earlier. Could you compare the accuracy with the existing methods?
Reviewer 3 Report
The authors present novel methods to calculate 3D skeletons from 2D detections. The proposed methods are very impressive, and the analysis results also exhibit very good performance. However, there appears to be a problem with collecting experimental data in this study, as the number of subjects is unknown, without the information about the approval of the institutional review board. The authors should give clear statements about these issues. In addition, other comments and errors are listed as follows.
- In the abstract, the authors claim that “The resulting RMS angle error is between 12° and 30°, showing that all three methods are accurate enough to calculate a useful REBA score from.” I think it is not the right statement because if the angle error reaches 30° that is unable to give a relatively accurate ergonomic assessment.
- In line 72, there has a typing error in the citation. Please revise it.
- Line 165, two repeated and identical verbs appear in a sentence. Please correct it.
- Section 4.4, is only one subject in this study? I think one subject is too few. Please explain why do the authors determine just to collect the data from one test subject?
- What is REBA? DoF? The full name of the abbreviation should appear at the first time shown in the article. Please correct the manuscript.
Round 2
Reviewer 1 Report
Thanks to the authors for answering all my questions. The quality of the revision is improved a lot. However, some critical concerns and important information are still lacking. Please consider the following commends for revising.
- Ln 20-22. “One of the most used methods is REBA [1] (Rapid Entire Body Assessment). Because most methods are similar, we will use REBA as an example use case.
-The reason for using REBA as an example is still weak. Moreover, what are the “most methods” mean? There are many MSDs assessments, and the differences between them are different. Each method is applied for a specific purpose. The author mentioned that “similar” is too subjective.
- Ln 22-23 “ However, the data generated by our techniques can be used for any ergonomic assessment that uses joint angles.”
-Do you verify this? How to confirm? Additionally, the statement is improperly added in the Introduction section. The readers do not sufficiently know what “your technique” stands for.
- The Discussion is still very weak. This part is a crucial part of a good paper. Please make more discussion based on your results.
- Only comparing the RMS results with one previous paper is not sufficient to support the proposed technique is good enough.
- The topic related to this paper has been a hot issue in recent years. Hence, some references were recommended in your revision to increase the quality and be applied to compassion. The authors can take these into consideration.
- An evaluation of posture recognition based on intelligent rapid entire body assessment system for determining musculoskeletal disorder
- Using the Microsoft Kinect™ to assess 3‐D shoulder kinematics during computer use.
- RGB‐D ergonomic assessment system of adopted working postures.
- Dose Figure 2 made by the authors? If not, please quote a reference.
Reviewer 2 Report
Thank you fo your response.
Author Response
No further response necessary.
Reviewer 3 Report
The authors have made necessary corrections and answered all questions in the revised manuscript.
Author Response
No further response necessary.